# Psychological and Biochemical Effects of an Online Pilates Intervention in Pregnant Women during COVID-19: A Randomized Pilot Study

**DOI:** 10.3390/ijerph191710931

**Published:** 2022-09-01

**Authors:** Hyun-Bin Kim, Ah-Hyun Hyun

**Affiliations:** 1Department of Biological Sciences, Daeduk University, 68, Gajeongbuk-ro, Yuseong-gu, Daejeon 34111, Korea; 2Department of Exercise Biochemistry and Exercise, Korea National Sport University, Seoul 05541, Korea

**Keywords:** pregnancy Pilates, online exercise, postpartum depression, body composition, serotonin

## Abstract

The purpose of this study was to analyze the effect of real-time online Pilates exercise during COVID-19 on women’s body composition, blood lipids, and psychological health after childbirth. The participants were 16 pregnant women (24–28 weeks pregnant) enrolled at the C Women’s Culture Center in Seoul, South Korea, classified into online Pilates groups and non-exercise groups (PE, n = 8; CON, n = 8). The online Pilates program was conducted for 8 weeks, twice a week, and 50 min a day using a real-time video chat app. Participants visited the hospital twice for body composition and blood tests. Questionnaires on postpartum depression, sleep disorder, and stress were conducted at 6 weeks and 12 weeks after childbirth. We found a significant difference between groups in body composition. The weight, percentage of body fat, body fat mass, and BMI of the PE group decreased. Blood lipids showed significant differences between the groups in TC, TG, LDL and CRP, while insulin and HDL showed no difference. All blood lipids, insulin, and CRP in the PE group were reduced. There were significant differences between the groups in postpartum depression, sleep disorders, and perceived stress indices performed in the post-test, and the serotonin concentration in the PE group increased. Serotonin levels were significantly correlated with postpartum depression, body fat mass, and body fat rate. Pregnant women’s online Pilates in this study was effective at reducing weight and depression in women after childbirth and should be used to promote women’s mental health during COVID-19.

## 1. Introduction

COVID-19 caused great social, economic, and cultural changes worldwide [1]. Many people experienced anxiety and stress due to the pandemic, especially pregnant women, who are at high risk due to the psychological effects of fetal protection [2]. The World Health Organization advised pregnant women with weak immunities not to go out, resulting in decreased physical activity, leading to obesity [3]. Obesity in pregnant women causes diabetes, high blood pressure, and maternal complications, which may cause fetal death [4]. In addition, prenatal care is important because obesity and depression during pregnancy are highly correlated and can cause serious mental illness after childbirth [4]. According to a recent meta-analysis, maternal depression increased by 25.6% and anxiety by 30.5% during the COVID-19 pandemic [5]. Furthermore, the American College of Obstetricians and Gynecologists (ACOG) warned that maternal anxiety had a negative effect on fetal growth and could cause premature birth, placenta previa, and underweight children [6]. Considering that “Corona Blue”, which means fatigue, lethargy, and depression caused by COVID-19, has been highly surveyed in women in their 30 s [7], continuous observation and management of pregnant women is warranted. In this regard, Cohen et al. (2020) developed guidelines for mental health support for pregnant women during COVID-19 [8]. Although several countries mention the importance of psychological support for mothers [9,10], alternatives are scarce, as national policies are concentrated on quarantine.

Pregnancy and childbirth are significant events for women. During this period, women experience considerable changes in the skeletal, circulatory, and reproductive systems [11]. Although changes in weight and body shape during pregnancy are natural, uncontrolled weight gain causes hyperlipidemia, chronic fatigue, back pain, and depression due to obesity [12]. Postpartum depression develops between six to eight weeks after delivery, with women complaining of emotional anxiety, lethargy, and depression [13]. Camisasca (2021) has reported that postpartum depression is significantly associated with parenting stress and insomnia [14], and pathological symptoms may persist for more than one year [15]. The Royal College of Obstetrics and Gynecology reported that because the COVID-19 environment could increase the negative emotions experienced by pregnant women, maternal care during the pandemic should be addressed at a national level [16]. In this regard, the ACOG recommends participating in prenatal education and care (meditating or participating in online exercises to maintain proper weight, for example) to foster the physical and emotional well-being of pregnant women [17].

Exercise has traditionally been used as a non-invasive treatment for obese and depressed patients, and its effectiveness has been demonstrated through several studies [18,19]. In particular, molecular mechanism studies on the anti-depression effects of exercise have suggested that exercise promotes tryptophan activation in the brain, increasing serotonin secretion, and reducing insulin resistance to prevent depression [20]. Serotonin, an appetite control hormone, also known as a “happiness hormone,” plays an important role in lipid metabolism and emotional control functions [21]. People with obesity have a lower serotonin concentration than people with normal weight, and a decrease in serotonin can promote appetite and cause depression [22]. According to previous studies, postpartum women’s parenting stress is caused by high calories, late-night meals, and alcohol, resulting in visceral fat accumulation and abdominal obesity [23]. Obesity caused by COVID-19 inhibits serotonin synthesis and this serotonin deficiency reduces brain nerve control function, promoting aging in women [24]. To address this problem, Cho (2016) reported that taking a walk in the sun for about 30 min a day promoted serotonin secretion [25]. Further, Wolf (2021) reported that patients with neuromuscular disorders who participated in regular exercise during COVID-19 experienced an alleviation of symptoms [26]. However, the effect of exercise during pregnancy on the emotional health of women after childbirth in an infectious disease environment is not evident in the literature.

Traditionally, exercise recommended for pregnant women includes walking, yoga, swimming, and ballet. Pilates has recently been reported to be effective in weight control and pain relief for pregnant women [11]. Pilates is an aerobic and anaerobic complex exercise that helps strengthen core muscle strength and correct unbalanced body shapes [27]. Particularly, Pilates in pregnant women reduces back and pelvic pain, and improves trunk muscles and overall balance, which helps prevent falls [28]. A study on the emotional effects of Pilates reported that Pilates using small tools increased pregnant women’s childbirth confidence, physical efficacy, quality of life, and helped improve physical strength for childbirth [29]. A study conducted during COVID-19 reported that Pilates in postpartum women with abdominal obesity was effective at reducing waist and hip circumference, and relieved insomnia and depression [30]. Furthermore, Ghram (2021) suggested that non-face-to-face exercise can have the same effect as face-to-face exercise [31], making it an appropriate alternative for obese or the elderly with mobility difficulties [32]. However, online exercise programs considering the characteristics of pregnant women are still sparse; hence, related solutions should be prepared.

In the unprecedented situation due to COVID-19, online exercise has expanded around the world, owing to highly developed technologies [33]. The American College of Sports Medicine (ACSM) recommends participating in home training or virtual sports, using virtual reality and augmented reality to prevent obesity and complications that have increased due to the pandemic [34]. Particularly, online exercise using real-time video apps is effective in motivating participants. Furthermore, obese employees who participated in tele-exercises showed reduced body fat, BMI, and visceral fat [35]. A study investigating the psychological effects of online exercise during COVID-19 found that doing physical activities lowers anxiety and promotes emotional well-being [36]. In addition, anxiety disorders and the quality of life of the elderly who participated in exergames improved [37]. However, to our knowledge, there is no clinical review that examines the physical and psychological effects of exercise during pregnancy on women after childbirth [38]. Therefore, building a new program that includes exercise intensity, time, and frequency for pregnant women will be an appropriate measure for maternal health in the post-COVID-19 era. Hence, the purpose of this study is to investigate the effects of pregnant women’s participation in real-time online Pilates on body composition, blood lipids, postpartum depression, sleep disorders, perceived stress, and serotonin after childbirth.

## 2. Materials and Methods

### 2.1. Study Design

The study was a randomized controlled experiment, involving 16 pregnant women participants. Participants were randomly chosen by a computer lottery and assigned to the Pilates exercise group (PE; *n* = 8) or the control group (non-exercise) (CON; *n* = 8). Assessment staff were blinded to the assignments. Randomized allocation concealment, and implementation was conducted by different staff. Our study was a pilot for verifying the effectiveness of non-face-to-face exercise. Our study complied with the recommendations of the Declaration of Helsinki. In addition, the Ethics Committee of the Korea National Sport University (1263-202109-br-018-01) provided approval for the study.

### 2.2. Participants

The participants were women registered at the ‘C’ Sports Center in Seoul, Korea, who understood the aim of the study and offered their voluntary participation. The selected participants included pregnant women under the age of 40 years who were at 24–28 weeks of single-fetus pregnancy. They did not have a diagnosis of diabetes or hypertension and did not take any medications. Furthermore, all participants had a BMI < 30, and participants with obesity or physical pain were excluded.

### 2.3. Online Pilates Exercise

All participants in the PE group participated in the exercise using a real-time remote chat application (ZOOM). The PE group used a personal computer or smartphone at home and exercised in real-time, while communicating with the trainer. The Pilates program consisted of a warm-up, main exercise, and cool-down. It was conducted for eight weeks, twice a week, 50 min a day. The exercise intensity was set to 50–60% of the maximum heart rate, which is the appropriate exercise intensity for pregnant women recommended by the ACOG (Figure 1) [6]. The intensity evaluation maintained the self-awareness exercise index (RPE) of 11 to 13 using Borg’s scale. Their RPE was monitored to maintain proper strength. The trainer observed the condition of the pregnant women during the sessions and gradually increased the intensity of exercise every three weeks, according to the fitness level (Table 1). Considering that the participant was a pregnant woman, precautions before movement were sufficiently explained, and in the case of complaints of physical discomfort during the exercise, they could stop the exercise and rest. The CON group did not participate in any exercise for eight weeks.

### 2.4. Body Composition Test

All participants were asked to stop food intake and empty their bladder two hours before the body composition measurement, according to the instructions. Height was measured using an automatic height scale (DS-103M, Dong Sahn Jenix Co., Seoul, Korea). After removing all metal accessories, weight (kg), body fat (kg), BMI (kg/m^2^), skeletal muscle mass (kg), and body fat percent (%) were measured using bioelectrical impedance analysis (In-Body 770, Biospace Co., Seoul, Korea). All participants wore light clothing and stood with their soles in contact with the foot and hand electrodes.

### 2.5. Blood Test

For the blood test, all participants fasted from 10 pm of the day before until 9 am on the day of the test. In addition, 10 mL of blood was drawn from the brachial vein for blood glucose, lipid, and serotonin tests. After 30 min of incubation at the laboratory’s temperature, blood was centrifuged (3000 rpm, five minutes) to separate the serum and was immediately taken to Green Cross Laboratories, Inc., to test the total triglycerides (TG), total cholesterol (TC), high density lipoprotein (HDL), low density lipoprotein (LDL), C-reactive protein (CRP), insulin, and serotonin. All the assays were carried out according to the instructions of the manufacturers.

### 2.6. Edinburgh Postnatal Depression Scale (EPDS)

We used the Edinburgh Postnatal Depression Scale to study postpartum depression of the participants. The validity of this scale has been verified for use among Korean mothers [39]. The EPDS is a self-report testing tool developed in the UK that tests postpartum depression by enquiring about depression, anxiety, and suicidal thoughts over the past week. The EPDS consists of ten items rated on a four-point Likert scale ranging from 0 to 3. The lowest total score that can be obtained is 0, whereas the highest is 30. Individuals who score 13 or more are considered at risk of developing depression. The reliability (Cronbach’s α) of the EPDS in this study was 0.83, and it was processed using SPSS 24.0 software (IBM, Armonk, NY, USA).

### 2.7. Pittsburgh Sleep Quality Index (PSQI)

We used the Pittsburgh Sleep Quality Index to study sleep quality in the participants after childbirth [40]. PSQI is a self-report questionnaire that evaluates sleep quality and disturbances over the past few weeks. It includes 18 items that are clustered together into 7 primary factors (sleep delay, sleep quality, sleep duration, sleep disability, habitual sleep efficiency, sleep medication use, and daytime dysfunction). Each item is scored from 0 to 3, and the total score ranges from 0 to 21. The total score is obtained by summing the scores of the seven factors. A higher total score indicates lower sleep quality. A total score of ≤5 indicates sound sleep, whereas a score of ≥8 indicates a sleep problem. The reliability (Cronbach’s α) of the PSQI in this study was 0.81.

### 2.8. Perceived Stress Scale (PSS)

The Perceived Stress Scale (PSS) was used to measure participant’s stress [41]. This 10-item tool has items about perceived stress in the past few weeks. Responses are scored on a five-point scale (0 = never, 1 = rarely, 2 = sometimes, 3 = frequently, 4 = very often). Negative items (nos. 4, 5, 7, and 8) are reverse scored. The total score is 40, with a higher score indicating greater severity of stress. The reliability (Cronbach’s α) of the PSS in this study was 0.83.

### 2.9. Statistical Analysis

All data were processed using SPSS 24.0 software. The differences in body composition, blood lipids, CRP, insulin, serotonin, EPDS, PSQI, and PSS between the PE and CON groups were analyzed. Due to the small sample size, nonparametric methods were used in the analyses. Independent t-tests were used to assess the differences in age, height, and weight between the Pilates and control groups. Pearson’s chi-square test was used to compare the frequencies of maternal BMI, smoking, and employment occupation between the PE and CON. Differences in the average changes (post-pre) between the two groups were analyzed using the Mann–Whitney U test and changes over time in each group were comparatively analyzed using the Wilcoxon signed rank test. A comparison of the reference values between groups was presented by Cohen’s d values. All statistical values were presented as mean and standard deviation, and the statistical significance was set at α < 0.05.

## 3. Results

### 3.1. Maternal Characteristics

A total of 16 women were randomized and 16 were excluded. Table 2 shows the general characteristics of the pregnant participants in the study groups. No significant differences (*p* > 0.05) in maternal characteristics were found between the groups at the baseline.

### 3.2. Effect of Pilates on Body Composition

After eight weeks of the experiment, the PE and CON groups showed significant differences in body weight (BW: z = −2.312, *p* = 0.021) (Table 3). The PE group showed significant changes in body fat mass (BFM), skeletal muscle mass (SMM), percent body fat (PBF), and BMI (BFM: z = −2.100. *p* = 0.036, SMM: z = −2.366 *p* = 0.018, PBF: z = −2.524. *p* = 0.012, BMI: z = −2.521. *p* = 0.012) (Table 3). The CON group showed significant changes in all measures except body fat mass (BFM: z = −1.863. *p* = 0.063, SMM: z = −2.524. *p* = 0.012, PBF: z = −2.521. *p* = 0.012, BMI: z = −2.521. *p* = 0.012) (Table 3).

### 3.3. Effect of Pilates on the Lipid Profiles

After eight weeks of experiment, the PE and CON groups showed significant differences in TC, TG, LDL, and CRP, whereas there were no differences in insulin or HDL (TC: z = −2.315, *p* = 0.002; TG: z = −2.836, *p* = 0.005; LDL: z = −2.735, *p* = 0.006; CRP: z = −2.432, *p* = 0.015) (Table 4). The PE group showed significant changes in all measures (TC: z = −2.521, *p* = 0.012; TG: z = −2.521, *p* = 0.012; LDL: z = −2.251, *p* = 0.012; HDL: z = −2.103, *p* = 0.035; insulin: z = −2.366, *p* = 0.018; CRP: z = −2.536, *p* = 0.011) (Table 4). The CON group showed significant changes in TC, LDL, HDL, and insulin (TC: z = −2.524, *p* = 0.012; LDL: z = −2.521, *p* = 0.012; HDL: z = −2.521, *p* = 0.012; insulin: z = −2.527, *p* = 0.012) (Table 4, Figure 2).

### 3.4. Effect of Pilates on the Psychological Effects

After eight weeks of experiment, the PE and CON groups showed significant differences in EPDS, PSQI, and PSS (EPDS: z = −3.138, *p* = 0.002; PSQI: z = −2.424, *p* = 0.015; PSS: z = −1.641, *p* = 0.001) (Table 5), but no differences in serotonin levels. The PE group showed significant changes in EPDS, PSQI, PSS, and serotonin (EPDS: z = −2.536, *p* = 0.011; PSQI: z = −2.640, *p* = 0.008; PSS: z = −2.539, *p* = 0.011, serotonin: z = −2.521, *p* = 0.012) (Table 5). The CON group showed significant changes in EPDS and PSS (EPDS: z = −2.449, *p* = 0.014; PSS: z = −2.401, *p* = 0.016) (Table 5, Figure 3).

## 4. Discussion

To address the rising obesity concerns caused by COVID-19, several experts have reported that participating in online exercise programs is effective [42]; however, evidence of the effectiveness of non-face-to-face exercise in pregnant women is limited. Additionally, there is a lack of studies that explore the effect of pregnant women’s exercise on obesity and psychological health of postpartum women during the COVID-19 pandemic. To address this gap, we conducted online Pilates for pregnant women. Our results that describe the women’s body composition, blood lipid, and postpartum psychological health are discussed as follows.

### 4.1. Effect of Pilates on Body Composition

Obesity during pregnancy has a negative physical and psychological effect [42], but there are very few studies on online exercise intervention during the COVID-19 pandemic. The ACOG recommends that pregnant women participate in appropriate levels of physical activity to avoid such stress [5,43]. Previous studies have reported that non-face-to-face exercise reduced systolic pressure, diastolic pressure, and fasting blood sugar, and increased quality of life in adult men and women [44]. A meta-analysis showed that eHealth interventions can prevent the risk of cardiovascular disease by reducing men’s weight, BMI, waist circumference, and blood pressure [45]. The results of the present study were similar, with weight, percentage of body fat, body fat mass, and BMI decreasing in the PE group. This is consistent with the studies that show that Pilates is effective at increasing women’s lower extremity muscle and grip strength. This also shows that online Pilates during COVID-19 is effective in relieving abdominal obesity [10]. Moreover, it is important to note that the period of intervention in this study was set differently from previous studies (effect of exercise in pregnant women and after childbirth). The weight loss in women after childbirth in this study suggests the importance of prenatal management during the COVID-19 pandemic and supports the findings of previous studies that non-face-to-face exercise has a similar effect as face-to-face exercise [30].

### 4.2. Effect of Pilates on the Lipid Profiles

Exercise and diet control are essential to quickly reduce increased blood lipids during pregnancy. In a related study, eight weeks of Pilates training reduced cholesterol and LDL in obese women [46]. The ACSM reported that 150 min or more of aerobic exercise per week is required to lose weight, and that the risk of adult disease can be reduced when combined with resistance exercise [32,33]. In this study, there were significant differences in TC, TG, LDL, and CRP between the PE group and the CON group. These results are consistent with the studies that show that real-time online high-intensity interval training in obese women (BMI > 25 kg/m^2^, and abdominal circumference > 85 cm) reduces visceral fat and LDL during COVID-19 [47], and is effective in preventing hyperlipidemia and diabetes [48]. Significant differences in CRP between groups in this study were confirmed and exercise during pregnancy was effective at relieving inflammation, recovering quickly, and reducing the weight of women after childbirth. On the other hand, there was no significant difference between the groups with insulin. These results are believed to be different to those that often appear in maternal exercise studies. Kampmann (2019) suggests that women during pregnancy or after childbirth have greater hormonal changes than the general population, and related blood lipid imbalances may appear [49]. Furthermore, considering the research results that women’s age, environment, job, and parenting level increase stress hormones after childbirth, concerns about newborns during COVID-19 may interfere with the function of insulin and appetite control hormones. However, it is meaningful that the participants of this study included women with similar hormone secretion (corresponding to 24–28 weeks of gestation). In addition, significant intergroup effects in blood lipids suggest that the exercise protocol of this study is suitable for online exercise intensity of pregnant women.

### 4.3. Psychological Effects of Pilates

Depression can negatively affect the quality of life and one’s daily life [50]. In particular, more than 80% of postpartum women experienced depressive symptoms, and parenting stress increased significantly after COVID-19 [5]. Although the pathophysiological mechanism for depression has not yet been fully identified, serotonin reduction and obesity have a high correlation, and the severity of the disease can be determined [21]. According to previous studies, exercise is recommended as a nonpharmacological treatment for depression. Moreover, serotonin and stress-related hormone functions are improved by directly stimulating brain neurotransmitters [19]. Studies also show that low intensity exercises, such as walking for less than 150 min per week, as well as moderate intensity exercises, can prevent depression [51]. In this study, there were significant differences between groups in EPDS, PSQI, and PSS of women after childbirth. However, there was no significant difference in serotonin between the two groups, but it increased significantly in the PE group. These results are consistent with studies that show that depressive symptoms improved in older adults who participated in underwater exercise [52], and online exercise helps reduce anxiety and stress during COVID-19 [53]. In addition, as a result of analyzing the association between the three psychological factors and serotonin, a high correlation with EPDS appeared, and the basis for the preceding study is presented (Table 5). Finally, there was a high correlation between body composition and serotonin in body fat mass and body fat rate (Table 6).

In summary, we found that exercise during pregnancy during the COVID-19 period alleviated weight and depression after childbirth and had a positive effect on serotonin levels. In addition, when observed by experts, online exercise was found to be safe and effective for pregnant women’s health. However, despite these positive results, this study has several limitations. First, this study cannot be generalized, due to the small number of participants. Second, we did not limit or modify the diet of the women. Since it was difficult to recruit pregnant women during the COVID-19 pandemic, we did not want to limit or modify their meals, which could potentially lead to increased stress. To compensate for this issue, the study participants were asked to write a daily diet diary, and a professional nutritionist checked it every Sunday. In future studies, a more specific and clear effect will be obtained if the effect of online exercise is verified with more participants. Online exercise has physical and emotional effects when personal characteristics are considered and implemented as real-time exercise. Therefore, the physical and psychological effects of online exercise conducted in this study present new exercise guidelines for pregnant women in infectious disease environments.

## 5. Conclusions

The purpose of this study was to investigate the effects of an eight-week real-time online Pilates program in pregnant women during COVID-19 on body composition, lipid metabolism, psychological health, and serotonin after childbirth. We found a significant difference between the groups in body composition in terms of weight, and the weight, percentage of body fat, body fat mass, and BMI of the PE group decreased. Blood lipids showed significant differences between the groups in TC, TG, LDL, and CRP, while insulin and HDL showed no differences. In the PE group, all blood lipids, insulin, and CRP were reduced. Finally, there were significant differences between the groups in depression, sleep disorders, and stress indices, and an increase in serotonin within the PE group. Therefore, real-time online Pilates with pregnant women, as examined in this study, is effective for weight control and depression relief for postpartum women and should be used as a tool to promote women’s mental health during the pandemic.

## Figures and Tables

**Figure 1 ijerph-19-10931-f001:**
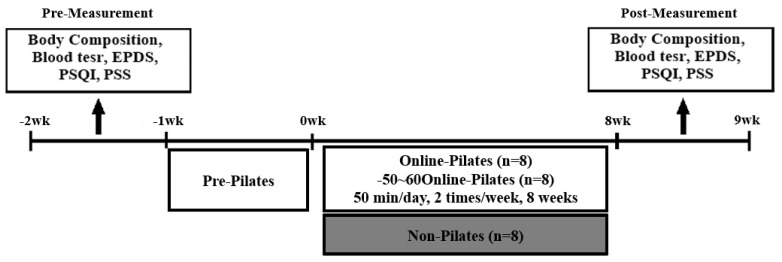
Experimental design. EPDS: Edinburgh Postnatal Depression Scale, PSQI: Pittsburgh Sleep Quality Index, PSS: Perceived Stress Scale, HRmax: maximal heart rate, RPE: rating of perceived exertion.

**Figure 2 ijerph-19-10931-f002:**
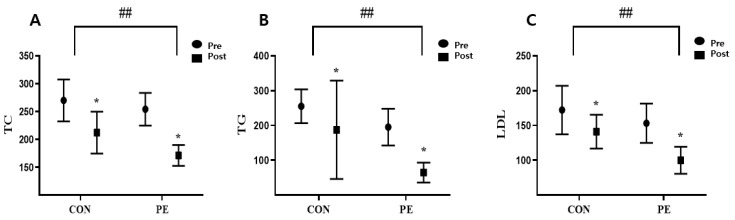
Effect of blood lipids profile according to Pilates. Bars represent mean ± SD (CON: *n* = 8, PE: *n* = 8). * *p* < 0.05 from pre to post. ^##^
*p* < 0.01 change (post–pre) between groups.

**Figure 3 ijerph-19-10931-f003:**
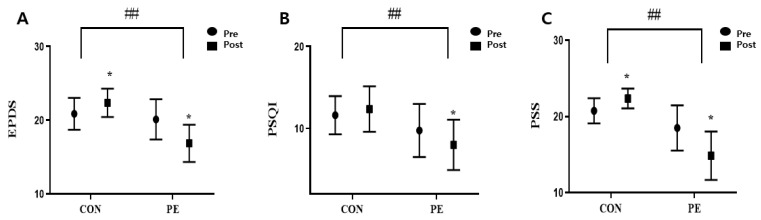
Effect of EPDS (**A**), PSQI (**B**), PSS (**C**) according to Pilates. Bars represent mean ± SD (CON: *n* = 8, PE: *n* = 8). * *p* < 0.05 from pre to post. ^##^
*p* < 0.01 change (post–pre) between groups.

**Table 1 ijerph-19-10931-t001:** Online Pilates program.

Modes	Contents	Time (min)	Reps, Set, and Rest	RPE
Warm-up	Soft stretching and breathing	5		10
Main exercise	Level 1: weeks 1–3 Half squat, cat cow, donkey kick, bridge, clam, spine rotation, arm circle, leg circles Level 2: weeks 4–6 Half-saw, half-lunges, hip hinge, side lateral raise, kneeing push-up, leg side up Level 3: weeks 7–8 Half-lunge twist, squat, low impact down dog, leg side kick, lunges, deep breathing	30	8–12 reps × 2 set 10 sec rest between sets	11–13
Cool-down	Deep breathing and total body stretching	5		10

**Table 2 ijerph-19-10931-t002:** Maternal characteristics at baseline.

Maternal Characteristics			
Variable	CON (*n* = 8)	PE (*n* = 8)	*p*
Age (years)	38.14 ± 1.39	39.71 ± 2.01	0.392
Maternal height (cm)	163.82 ± 3.71	164.81 ± 4.43	0.548
Maternal weight (kg)	64.55 ± 2.52	62.71 ± 4.00	0.135
Maternal BMI (*n*/%)	22.88 ± 1.64	24.25± 1.58	0.584
22–25	6/75	6/75	
25–28	2/25	2/25	0.228
Smoking			0.388
No	7/87.5	6/75	
Yes	1/12.5	2/25	
Occupation (*n*/%)			0.294
Sedentary job	3/37.5	1/12.5	
No job	5/62.5	7/87.5	

**Table 3 ijerph-19-10931-t003:** Body composition.

	CON (*n* = 8)		PE (*n* = 8)		Diff (Post–Pre)	
	Pre	Post	Pre	Post	*p*	Cohen’s d
SMM (kg)	23.50 (1.30)	21.11 (1.47) *	21.06 (1.34)	19.67 (1.00) *	0.058	1.145
BFM (kg)	21.02 (3.61)	20.33 (2.96)	22.22 (2.37)	19.45 (4.14) *	0.753	0.244
PBF (%)^)^	32.47 (4.57)	34.03 (4.06) *	35.92 (2.96)	33.15 (4.56) *	0.793	0.203
BMI (kg/m^2^)	24.81 (1.60)	22.90 (1.44) *	23.33 (1.58)	21.58 (1.57) *	0.083	0.876

Values are presented as mean ± SD (*n* = 8 per group). Main time effect: * *p* < 0.05, pre- versus post-Pilates period in the within groups. CON: non-Pilates exercise, PE: Pilates exercise, SMM: skeletal muscle mass, BFM: body fat mass, PBF: percentage of body fat, BMI: body mass index.

**Table 4 ijerph-19-10931-t004:** Blood lipids.

	CON (*n* = 8)		PE (*n* = 8)		Diff (Post–Pre)	
	Pre	Post	Pre	Post	*p*	Cohen’s d
TC	270.25 (37.62)	212.25 (37.62) *	254.25 (29.45)	171.25 (18.75) *	0.002 *	1.379
TG	255.25 (48.85)	187.62 (141.27)	195.50 (52.80)	65.00 (28.60) *	0.005 *	1.203
LDL	172.25 (34.78)	141.25 (24.39) *	153.37 (28.18)	100.00 (19.47) *	0.006 *	1.869
HDL	82.87 (16.11)	65.25 (10.47) *	86.50 (11.27)	73.25 (15.86) *	0.916	0.595
Insulin	7.11 (0.77)	4.91 (1.15) *	7.13 (0.79)	3.93 (1.04) *	0.091	0.893
CRP	1.96 (1.56)	1.32 (1.14)	0.77 (0.25)	0.25 (0.11) *	0.015 *	1.321

Values are presented as mean ± SD (*n* = 8 per group). Main time effect: * *p* < 0.05, pre- versus post-Pilates period in the within groups. CON: non-Pilates exercise, PE: Pilates exercise, TC: total cholesterol, TG: total triglycerides, HDL: high density lipoprotein, LDL: low density lipoprotein, CRP: C-reactive protein.

**Table 5 ijerph-19-10931-t005:** Psychological effects.

	CON (*n* = 8)		PE (*n* = 8)		Diff (Post–Pre)	
	Pre	Post	Pre	Post	*p*	Cohen’s d
EPDS	20.87 (2.16)	22.37 (1.93) *	20.12 (2.74)	16.87 (2.53) *	0.002 *	2.444
PSQI	11.62 (2.32)	12.37 (2.77)	9.75 (3.24)	8.00 (3.07) *	0.015 *	1.494
PSS	20.75 (1.66)	22.37 (1.30) *	18.50 (2.97)	14.87 (3.18) *	0.001 *	3.087
ST	83.66 (31.83)	88.48 (34.41)	94.13 (39.45)	140.86 (64.13)	0.115	1.017

Values are presented as mean ± SD (*n* = 8 per group). Main time effect: * *p* < 0.05, pre- versus post-Pilates period in the within groups. CON: non-Pilates exercise, PE: Pilates exercise, EPDS: Edinburgh Postnatal Depression Scale, PSQI: Pittsburgh Sleep Quality Index, PSS: Perceived Stress Scale, ST: serotonin.

**Table 6 ijerph-19-10931-t006:** Correlations between body composition and psychological effects and serotonin.

		BW	SMM	BFM	PBF	BMI	EPDS	PSQI	PSS
Serotonin	*p*	−0.143	0.448	−0.670 **	−0.699 **	0.033	−0.627 **	−0.338	−0.361
		0.597	0.082	0.005	0.003	0.905	0.009	0.200	0.170

BW: body weight, SMM: skeletal muscle mass, BFM: body fat mass, PBF: percentage of body fat, BMI: body mass index, EPDS: Edinburgh Postnatal Depression Scale, PSQI: Pittsburgh Sleep Quality Index, PSS: Perceived Stress Scale. ** *p* < 0.01.

## Data Availability

Not applicable.

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
