# Peer review of "Psychological and Biochemical Effects of an Online Pilates Intervention in Pregnant Women during COVID-19: A Randomized Pilot Study"

_ijerph, 2022, doi:10.3390/ijerph191710931_

Round 1
Reviewer 1 Report (Previous Reviewer 4)
The authors have addressed some of the comments from the reviewers. I think that especially the figures have been improved from the former version. While I had also provided some specific comments regarding the different sections in the manuscript, as stated before, my main concern regarding this work is the short sample size. Unfortunately, this is a methodological weakness and, under my point of view, the only way to overcome this limitation is to enlarge the sample size by keeping recruiting women for the program.
Author Response
Please see the attachment.

Reviewer 2 Report (Previous Reviewer 2)
1. This is a randomized controlled experiment. The sample size estimation or power analysis should be added. It is better if authors could provide the model format they want to establish.
2. In table 2, body weight is not necessary, because it is closely related to height. BMI is a comprehensive index.
3. In this study, statistical analysis did not adjust for factors such as sociodemographic characteristics, lifestyles, dietary patterns, etc. Thus, the difference may be not necessarily caused by Pilates, and further study is required. The authors should carefully consider whether conclusion can be made to emphasize the effect of online Pilates on the ‘prevention’ before participants suffered from depression and the ‘treatment’ after participants suffered depression in women after childbirth.
Author Response
Please see the attachment.

Reviewer 3 Report (Previous Reviewer 1)
I accepted the revised above manuscript and that the manuscript has been significantly improved and now warrants publication in ijerph.
Author Response
Please see the attachment.

This manuscript is a resubmission of an earlier submission. The following is a list of the peer review reports and author responses from that submission.
Round 1
Reviewer 1 Report
This is a nice study. However, I do not see why such a small of participants (N=16) in such a study employed a quantitative design/ statistical analysis.
Also, one would expect the authors to discuss the study's results and compare and contrast them with previous studies. Instead, this study discussion merely confused on highlighting the importance of the study, as it would be in the introduction section. So, it, means that the discussion section was merely presenting the results.
Author Response
Thank you for your precious time.
I am sending you a reply to your comment.
I marked the revised part in yellow in the text.
Thank you again.
Please see the attachment

Reviewer 2 Report
This paper focus on psychological and biochemical effects of an online Pilates intervention in pregnant women during COVID-19. It is a topic of interest to the researchers in the related areas; however, some remaining questions emerge.
1. Introduction---Too long and a run-on. For example, the first two paragraphs were about the effects of COVID-19 on health in pregnant women. The main outcome of this study was effects of online Pilates intervention on health in pregnant women.
2. Statistical analyses---This is a randomized controlled experiment. Please add the sample size estimation or power analysis. It is better if authors could provide the model format they want to establish.
3. Results---In table 2, body weight is not necessary, because it is closely related to height. BMI is a comprehensive index.
4. Results---In table 2-4, differences between CON and PE of baseline data should be analyzed. Then, differences of the change between post-data and pre-data in CON and PE should be performed.
5. Results---The data was presented as tables and figures, but the table and figure should not be repeated. If there is repetition, please select one presentation (e.g. table 3 & figure 1; table 4 & figure 2).
6. Results--- In table 2-4, statistical analysis did not adjust for factors such as sociodemographic characteristics, lifestyles, dietary patterns, etc. Thus, the difference may be not necessarily caused by Pilates, and further study is required.
Author Response
Thank you for your valuable time.
Submit a reply to your comment.
I marked the revised contents in yellow.
Thank you again.

Reviewer 3 Report
The paper entitled “Psychological and Biochemical Effects of an Online Pilates Intervention in Pregnant Women During COVID-19” addresses an interesting and relevant topic. However, I do have a major concern and some other minor points to discuss. Please see some comments below, which I think might help to improve the quality of the manuscript.
Major concern
The authors said that this is a pilot study: “Our study was a pilot for verifying the effectiveness of non-face-to-face exercise.” (lines 121-122). As such, this information must be included in the title of the paper and the aims/hypothesis of the study should be modified. The goal of a pilot work is not to test hypotheses about the effects of an intervention (in this case, online Pilates exercise), but rather, to assess the feasibility/acceptability of an approach to be used in a larger scale study.
If that was a mistake/typo on the manuscript and this experiment was not a ‘pilot study’, then the small sample sizes used here do not seem appropriately powered to answer questions about efficacy. The authors in this case must report on the paper the information related to their power calculation for this study.
Minor concerns
Abstract
“There were significant differences between groups”, “were significantly correlated”, these statements seem vague and do not add useful information to the abstract.
Keep the terms consistent: the ‘Pilates Exercise’ group is referred to as ‘online Pilates’ at the conclusion of the abstract.
Introduction
Reference number 4 (line 34) does not seem to be the most appropriate for this sentence. Same for reference number 11 (line 50).
Methods
This is a pilot study and this information should be reflected on the manuscript title.
As such, the goal of pilot work is not to test hypotheses about the effects of an intervention (in this case, online Pilates exercise), but rather, to assess the feasibility/acceptability of an approach to be used in a larger scale study. Thus, this pilot study should not be answering the question “Does this intervention work?” Instead this study should address the questions “Can we do this?”
With this in mind, I suggest the authors re-think the aims and hypothesis of the paper.
Was the Borg’s scale applied online? Has this been validated? Why not send them heart rate monitors?
Results
Please consider labelling the figures with A, B, C etc where appropriate.
Revise the vertical axis values as some of the error bars were off the chart.
None of the charts seem to start at the 0 point on the y axis.
The white box ‘pre’ and block box ‘post’ does not match what the figure shows (circles and squares)
Fig 1 – TC has some dots/arrows/signs below the squares. Please explain what do they main in the figure legend.
The figures are not clear. Why did the authors decide to show, what I assume is, CON pre vs post and then PE pre vs post, instead of CON pre vs PE pre, then CON post vs PE post?
Discussion
Lines 252-257 - This part of the discussion is not related to the data presented, rather it is re-stating information found in the introduction/background.
Avoid using ‘normal levels’ (line 253). Maybe change it to ‘pre-pregnancy’.
The sentence about COVID-19 is ambiguous (line 255-256). Do the authors mean that COVID-19 directly caused rapid weight gain or perhaps these would be indirect effects?
In many instances, the discussion only says ‘significant differences…’ without actually saying if it was a decrease or increase in the parameters or timing (pre/post) being discussed.
Line 291-293, ‘similar hormone secretion’ is a vague term and it was not measured. It seems like this would be more accurate if the authors refer to their inclusion criteria instead.
Table 5 – this should be moved up to the results section. It also seems like it was not referred to until the discussion.
Author Response

(The authors gave the same response as above.)

Reviewer 4 Report
The authors address an interesting research question: How does a Pilates program affect pregnant women health? However, it presents important limitations. The first and most important one is the sample size, 16 women is an excessively short number to make inferences from the study.
Below there are some other considerations about the different sections of the work.
Introduction.
Some more specific information regarding previous evidence on exercise programs on pregnant women is missing. There is an important research line which has addressed the effect of exercise on pregnant women health and this is not reflected in this section.
Methods
The procedure is not clear. I would recommend the authors to include a diagram-figure to display the length of the study as well as the collection data points specifying which specific measures were assessed at each time.
Some more information on the control group should be provided. Were these women active?
Regarding format, section 2.3 appears several times
Results
The mean scores and differences should be presented more clearly. This section should allow the reader to see differences intergroups before and after the intervention, as well as intragroup differences between pre and post. In the current state, it is not clear what information is provided in the last columns of the means differences tables.
The figures are not clear and the legends cannot be understood. Also, the name of the Tables should be improved so that they can more accurately describe what information the tables display.
Title of the section 3.1 should be corrected, since the word effects appears twice
Discussion
The discussion do not cite relevant works which have developed similar interventions among pregnant women. There are some paragraphs providing information which might be more suitable for the introduction (e.g., lines 252-261, 297-306)
Author Response

(The authors gave the same response as above.)

Round 2
Reviewer 2 Report
In the revised manuscript, I don’t find a revision to questions in the version 1. The questions in the version 1 are as follows:
“This paper focus on psychological and biochemical effects of an online Pilates intervention in pregnant women during COVID-19. It is a topic of interest to the researchers in the related areas; however, some remaining questions emerge.
1. Introduction---Too long and a run-on. For example, the first two paragraphs were about the effects of COVID-19 on health in pregnant women. The main outcome of this study was effects of online Pilates intervention on health in pregnant women.
2. Statistical analyses---This is a randomized controlled experiment. Please add the sample size estimation or power analysis. It is better if authors could provide the model format they want to establish.
3. Results---In table 2, body weight is not necessary, because it is closely related to height. BMI is a comprehensive index.
4. Results---In table 2-4, differences between CON and PE of baseline data should be analyzed. Then, differences of the change between post-data and pre-data in CON and PE should be performed.
5. Results---The data was presented as tables and figures, but the table and figure should not be repeated. If there is repetition, please select one presentation (e.g. table 3 & figure 1; table 4 & figure 2).
6. Results--- In table 2-4, statistical analysis did not adjust for factors such as sociodemographic characteristics, lifestyles, dietary patterns, etc. Thus, the difference may be not necessarily caused by Pilates, and further study is required.”
Reviewer 4 Report
I appreciate the authors' response to my previous coments and congratulate them for the development of the figure. However, I still find worrying the short sample size.